# Combining Artificial Neural Networks and GIS Fundamentals for Coastal Erosion Prediction Modeling

**Angeliki Peponi [1,2,3,\*]**, **Paulo Morgado [1,3]** and **Jorge Trindade [3,4]**

1   Geomodlab, Institute of Geography and Spatial Planning, Universidade de Lisboa,
    Rua Branca Edmée Marques, 1600-276 Lisboa, Portugal; paulo@campus.ul.pt
2   Faculty of Environmental Sciences, Czech University of Life Sciences Prague, Kamýcká 129,
    Praha-Suchdol 16500, Czechia
3   Centre of Geographical Studies, Universidade de Lisboa; Rua Branca Edmée Marques, 1600-276 Lisboa,
    Portugal; jorgetrd@campus.ul.pt
4   Department of Sciences and Technology, Universidade Aberta, 1269-001 Lisboa, Portugal
*   Correspondence: a.peponi@campus.ul.pt; Tel.: +351-933-606-757

**Abstract:** The complexities of coupled environmental and human systems across the space and time of fragile systems challenge new data-driven methodologies. Combining geographic information systems (GIS) and artificial neural networks (ANN) allows us to design a model that forecasts the erosion changes in Costa da Caparica, Lisbon, Portugal, for 2021, with a high accuracy level. The GIS–ANN model proves to be a powerful tool, as it analyzes and provides the "where" and the "why" dynamics that have happened or will happen in the future. According to the literature, ANNs present noteworthy advantages compared to the other methods that are used for prediction and decision making in urban coastal areas. In order to conduct a sensitivity analysis on natural and social forces, as well as dynamic relations in the dune–beach system of the study area, two types of ANNs were tested on a GIS environment: radial basis function (RBF) and multilayer perceptron (MLP). The GIS–ANN model helps to understand the factors that impact coastal erosion changes, and the importance of having an intelligent environmental decision support system to address these risks. This quantitative knowledge of the erosion changes and the analytical map-based frame are essential for an integrated management of the area and the establishment of pro-sustainability policies.

**Keywords:** geographic information systems; artificial neural networks; backpropagation; coastal urban zones; erosion changes prediction

## 1. Introduction

Coastal zones (CZs) provide valuable habitat and ecosystem services along with valuable perspectives on economic growth. The land-use and land-cover (LULC) dynamics in CZs are changing due to the increase of inhabitation and urbanization, which is largely due to the growth of the tourism industry and leisure activities, as well as sea-level rise [1,2]. World coasts, deltas, and estuaries presently support a large portion of the planet's population, and are likely to host a growing number of inhabitants in the near future, as most of the world's cities and megacities are located in the coastal zone [3–8]. Consequently, sea-level rise is among the major drivers for the population exposure to natural hazards, increasing its vulnerability to coastal erosion, flooding, arable land loss, and potable water shortage. Changes in costal LULC can trigger coastal erosion through direct interference on the sediment transport cycles and pathways [9–11]. The usual answers to cope with these issues involve hard and/or soft protection defense measures that mitigate coastal erosion at specific sites and at

micro levels [12,13]. However, an integrated coastal zone management approach that helps national and regional stakeholders and local communities deal with coastal erosion while benefiting from the coastal sustainable economic opportunities is presently lacking. Therefore, coastal erosion constitutes a socioeconomic problem locally, and also has regional effects through potential considerable financial losses and environmental degradation [14].

Comparing shorelines and coastlines in different time periods, we can identify and measure coastal erosion when the shorelines shift landwards between two time frames [15], as cited in [16,17]. Identifying and monitoring the changes in the spatiotemporal shoreline position facilitates the recognition of the ways that the CZs react to human activities and natural processes.

The relationships between humans and the environment and the hierarchical structures are characterized by nonlinearity; the underlying processes are either unknown or hard to understand. Several approaches to quantify human–erosion interaction have been proposed in recent years, including hazard/vulnerability exposures and dependencies [18–23]. These methods are mainly physical-based and lack anthropogenic variables. Moreover, they are relying on a linear type of method for the analysis of the variables, which means that variables are pre-worked out and classified.

The aim of this paper is to provide a robust tool combining geographic information systems (GIS) and artificial neural networks (ANNs) in order to predict the areas, within the study area, that are more prone to erosion in the near future (year 2021), by taking into account the driving factors (i.e. driving independent variables) upon the erosion phenomena (i.e. the dependent variable). One of the most important attributes of ANNs is that they can adjust to periodic changes, and identify patterns in complex natural nonlinear systems as the CZs. Our GIS–ANN model is nonlinear and therefore data-driven, which means that it has the ability to incorporate the uncertainty of complex systems, such as coastal erosion. Understanding the relationships between the social and natural forces and forecasting future erosion trends could be a vital tool for coastal management, decision making, land-use planning, and sustainable development [24]. Although, to really understand and explain the results highlighted by ANN, the analysis of the mechanical processes is the key, as local factors seem to be the root of such dynamics.

## 2. Materials and Methods

### 2.1. Study Area

The Portuguese western Atlantic coastal zone is characterized by high wave energy during the winter [25], where rocky coastal systems are predominant. However, locally sandy systems occur, varying in extent between sub-kilometric lengths and tens of kilometres.

The Costa da Caparica coastal zone is part of a coastal cell that extends from the Tagus river estuary to the Cabo Espichel, which is over almost 30 km, and is mainly characterized by low altitude/slope sandy beach–dune systems [23]. Morphologically, the study area has an arc shape with an orientation south/southeast and north/northwest (SSE–NNW). The study area represents an extensive coastal plain along the shoreline up to a fossil cliff. It is situated on a terrace about 50 meters above the mean sea level, and travels 4.5 kilometers from the shoreline to the edge of the cliff, totaling a 45 square kilometers surface [26]. The Caparica fossil cliff is located south of the Costa Caparica town, and extends to the Albufeira lagoon. The cliff is composed of unconsolidated fluvial sediments from the Pliocene correlated to the onshore Tagus paleovalley. As it is a tremendously important feature; in 1884, the area was characterized as a protected area [27]. The slope is gradual, and the bathymetric lines are nearly parallel to the shoreline.

Geologically, the study area consists mainly of beach and dune deposits whose distribution depends on wave/coastal drift circulation patterns over time, as well as Tagus River inputs [28]. Storms and floods influence the amount of sedimentation in the estuary, and therefore on the availability of sediments on the Caparica coastal system [27]. The coastal plain, due to its low topography and the fragility of the sand dune system, is vulnerable to flooding [29] and erosion [30]. The Caparica

coastal system is highly affected by erosion processes [30–32]. In response to this issue, a series of hard structures were built in the 1970 and 1980s of the past century to protect local communities from coastline retreat. These nine protection groins and seawalls prove to be insufficient because today, there is a permanent need for hard structure maintenance and artificial beach-sand nourishment. High-energy waves and the sediment drift place the study area as one of the most exposed to erosion and floods in Portugal [12]. The general swell period of the waves in the study area is between three and 16 s with local medium spectral directions of S-20-W to WNW. The most frequent local waves have directions of 260º to 290º, heights of 0.5 m to 2.5 m, and a period of five to eight seconds. In the study area, semi-diurnal tides occur on an approximate 12-h 25-min tide cycle, which is from south to north. The velocities of the tides in the Tagus estuary are intense, with low heights. They go beyond 2.0 m/s during the flood, and 1.8 m/s during the ebb, in the spring. The average values are 1.5 m/s and 1.4 m/s, respectively. In Costa Caparica near the river inlet, the tides are less strong, with values of 0.2 m/s and the directions of south to north due to the closed circuit [10].

The study area includes the two villages, Trafaria and Caparica, which belong to the municipality of Almada, Lisbon district, Portugal. The study area is located on the southern bank of the Tagus River (Figure 1). Costa de Caparica is an important second-home urban center with nearly 14,000 inhabitants [6]; it constitutes a major destination for holidays and surfing due to its high quality water and sand [7], as well as its location. The economic value of the waves in Costa de Caparica is between 50,000 to 1,000,000 Euros, considering an estimation of 22,000 surfers per year [8].

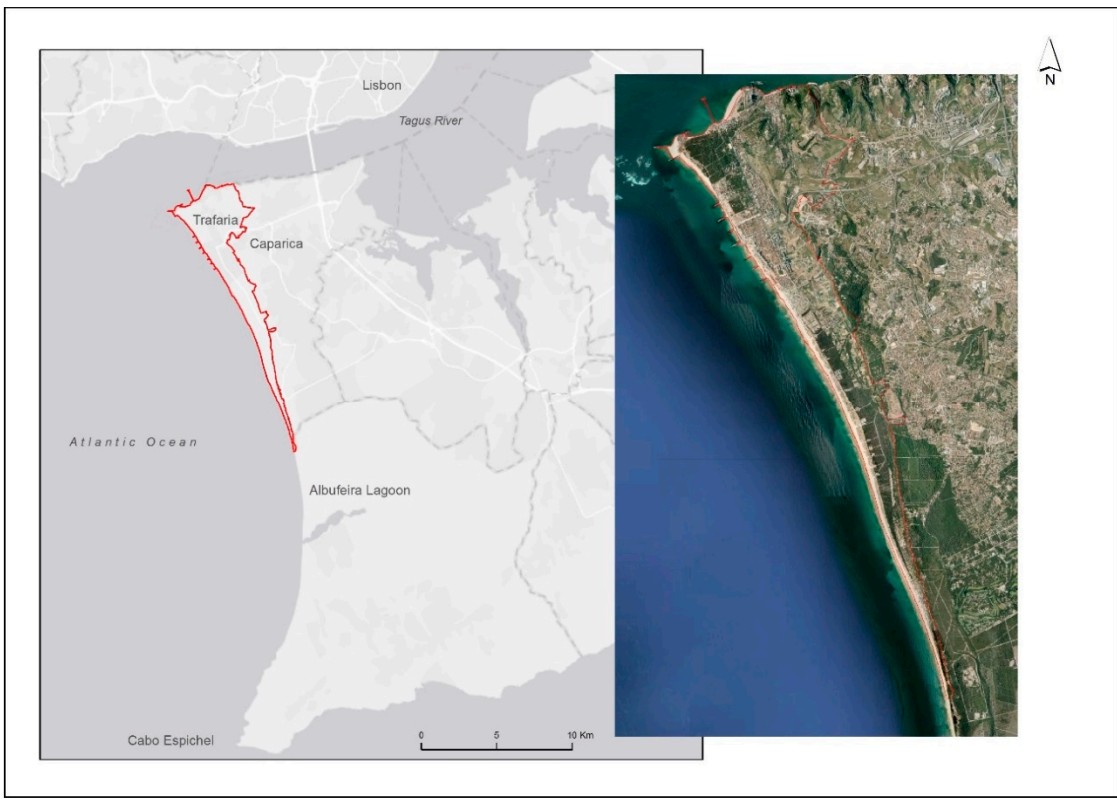

**Figure 1.** Location of the Study Area.

It is necessary to manage and protect the geomorphological and ecological attributes of the coastal systems in order to preserve them [14].

*2.2. Data*

Social/demographic and environmental data have been used for this study "I", which were acquired from the 2001 and 2011 census data found on the Portuguese National Statistical Institute

(INA) and from satellite images. Furthermore, Corine land cover for the years 2000 and 2006 has been used. We identified coastal erosion changes ("Area of erosion plots") over time between the years 1967, 1980, 1995, and 2008, by comparing the different shoreline positions for these years, which were taken from the master thesis of Sousa (2015) [30]. Spots of coastline landwards displacement were collected in a geodatabase using the dune vegetation line from historical data based on orthorectified aerial photographs and orthophotograph maps, which were the main proxi for coastline positions [30,33–38]. Nowadays, alterations of the vegetation in dunes are caused mostly by human activities which destabilize them and make them prone to erosion processes [39]. In order to identify the degree of human pressure to dune vegetation, satellite images have been classified into two categories: vegetated (gridcode 1) and non-vegetated (gridcode 0) using Iso Cluster Unsupervised Classification in GIS. The variables used for the study are listed in Table 1.

**Table 1.** Legend of Variables.

| Code | Name | Units |
| --- | --- | --- |
| RESID01_ | Resident population—2001 | Nº residents |
| RESID11_ | Resident population—2011 | Nº residents |
| CORNL00_ | Corine Land Cover—2000 | Level 1 CLC |
| CORN06_ | Corine Land Cover—2006 | Level 1 CLC |
| ALOJ01_3 | Households—2001 | Nº of households |
| ALOJ11_3 | Households—2011 | Nº of households |
| ER6080_R | Area of erosion plots—1967–1980 | Sq m |
| ER8090_R | Area of erosion plots—1980–1995 | Sq m |
| ER2008_R | Area of erosion plots—1995–2008 | Sq m |
| VEG_RCID | Area of vegetated and non-vegetated plots—currently | Sq m |

## 2.3. Methods

Amongst all the developed information technologies, GIS is considered an excellent tool to examine spatial distributions and therefore contribute to spatial decision making. GIS offers a complete and precise information technology platform, with large database storage power, advanced spatial analysis capability, and computer graphic processing. The GIS revolution of the mid-1980s was characterized by innovation in the collection, storage, manipulation, and management of geographical information [40]. GIS developments are still in progress, which means that most of the available analysis and modeling tools are pre-GIS. Only recently, in the first decade of the 21st century, did GIS start to integrate some smart tools, and therefore feature new algorithms that are capable of dealing with complex problems by making good use of all the available geographical information [40,41]. A big contribution to the GIS revolution is the geospatial policies of interoperability [42], which makes data migration between GIS and data analysis software possible, guaranteeing data integrity. Nonetheless, machine learning algorithms such as artificial neural networks or deep learning are still scarce in the GIS software packages. The availability of sparse and partial data and the limitation of a more knowledge-based approach make ANN a useful tool, thanks to its main characteristics such as freedom of assumptions, inherent nonlinearity, and its ability to deal with noise data in difficult non-ideal contexts, such as that in our case study [43]. Thus, in this project, GIS and ANN were combined to build and implement an accurate model of coastal erosion prediction and explore the potential of ANN.

Different types of ANN architecture have been compared by many researchers to prove their capability for a wide range of applications, and have been used as a tool for decision making in various published papers and textbooks [44–48]. In this study, the radial basis function network (RBF) and multilayer perceptron (MLP) have been tested with different network topologies and training algorithms to predict the shoreline movements in the near future and conduct sensitivity analyses on natural and social forces, and dynamic relations in the dune–beach system of Costa da Caparica, Lisbon, Portugal.

Artificial Neural Networks

Artificial neural networks (ANN) or simply a neural network, is a set of independent neurons linked together in the same way as the synapses, neurons, and dendrites of ours brain. Neurons are excited through these links, similar to electric signals, by the input values and by other neurons, propagating the excitation toward an output. In order for a neural network to learn and therefore execute some task, it must be trained. During the training, the network modifies the weights of the links among the neurons in a way that each input produces the expected output [49]. According to the way that the machine learns, there are two different types of ANN architectures: the supervised and non-supervised. In a supervised network, we choose the output (dependent variable) and provide a classified input data to the network; then, a learning function trains a data sample based on the given input data, so that each output value is instructed on what the expected input response signals should be. In a non-supervised network, we don't need to choose the output because the network organizes the input data in clusters or in categories, seeking similarity between the data; it learns from the main properties of the data, and then projects the information into an output map. In our paper, we chose the supervised ANN, which is the most useful [40], as well the most common form of machine learning, whether deep or not [50]. RBF and MLP are two of the most commonly used supervised artificial neural networks methods.

The radial basis function network (RBF) came into view as a variation of ANN in the late 1980s [51]; it has a three-layer feedforward architecture including an input layer, a hidden layer, and one output layer (Figure 2). The neurons of the input layer distribute the information to the hidden layer nonlinearly [47,52]. Each hidden layer node constitutes a radial basis activation function with a related center position and width. The output of each RBF depends on the radial distance between the introduced input pattern and the center position of each hidden node. A greater output is produced when this distance decreases [53–55]. The centers and the widths of the RBF units are defined either arbitrarily or by using clustering algorithms, such as for example k-means algorithms [56]. Afterward, a least mean square algorithm optimizes the weights [57]. The outputs of the hidden nodes are connected directly to the elements of the output layer. The output nodes perform a weighted sum of the outputs of the hidden layer using a linear activation function.

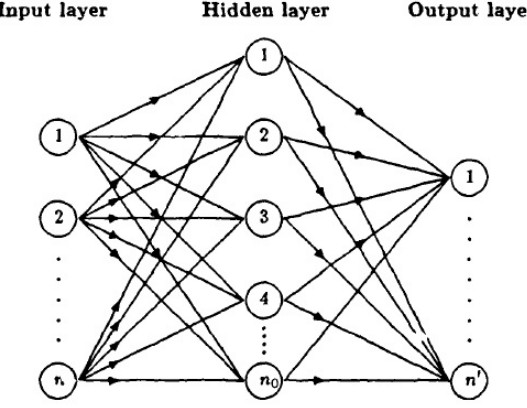

**Figure 2.** Schematic diagram of the feed-forward layered network model presented by the radial basis function (RBF) expansion [51].

Multilayer perceptron (MLP) it is more common type of network that is used for both pattern recognition [50] and function approximation [58,59]. The MLP consists of a three-layer feedforward neural network; it contains an input layer, the hidden layer, and an output layer. The neurons in the input layer transmit data to the next layer, while the neurons in the hidden and output layers are processing the data. The neurons in the input layer connect to a subsequent hidden and then output layer, and provide weighted inputs to which every neuron in the hidden and output layers respond [60]. Rumelhart and McClelland (1986) introduced the backpropagation (BP) learning rule

for MLP. The BP is one of the most useful training algorithms, and the most appropriate for training MLP. One of the reasons is that it requires low memory and can very rapidly approach an acceptable error. BP constitutes an algorithm with a three-stage procedure. Initially, the network starts with small random network parameter values. An input vector sample is introduced to the network and propagated forward to define the signals of the output layer. In the second phase, the output vector is compared with the desired outputs that are present in the dataset, resulting in error an signal that is spreading backwards to the hidden layer in the third stage. The error between the true and target outcome is used to shift the weights in the network so that the error has a higher probability of being lower. The forward and backward passes are repeated until the network is satisfactorily trained for the attributes of the targeted utility [61], and the overall error throughout the entire training set is reduced. Backpropagation is using a gradient descent method [62] to find the minimum of the output error function.

*2.4. Model Implementation*

To implement our ANN–GIS combined model, we have used a threefold technology including ArcGIS, STATISTICA, and IDRISI (Figure 3). ArcGIS was used to organize the data in a geodatabase and visualize the output results in a map form. To pass the features from the geodatabase to STATISTICA and run the ANN algorithms, we used IDRISI, since it guaranties data integrity as migrating the data to STATISTICA.

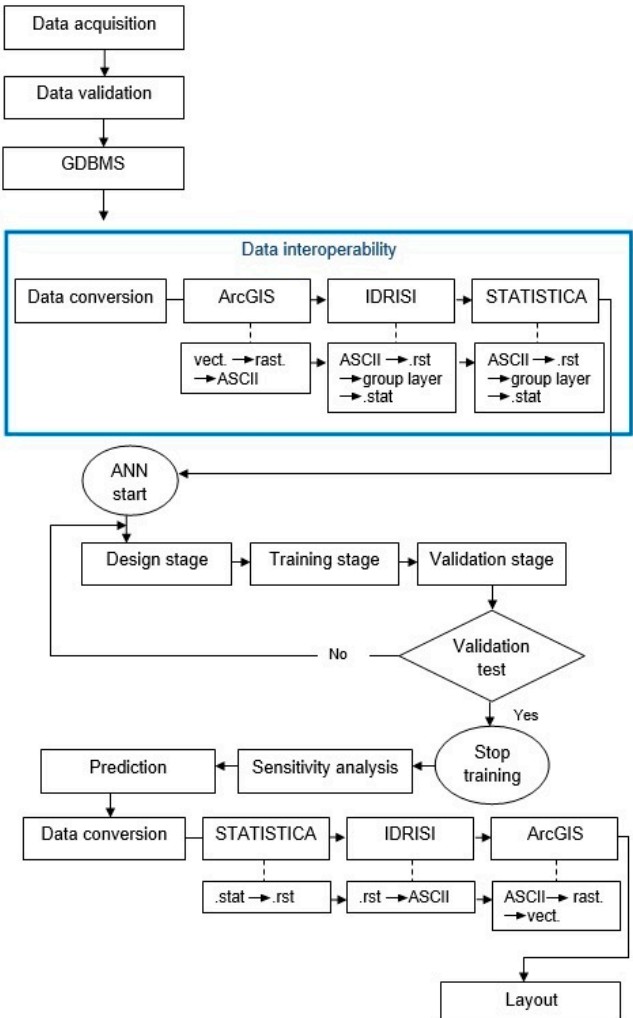

**Figure 3.** Physical Model of the Current Study [authors].

In STATISTICA, we ran the data using RBF and MLP types of neural networks. Comparing the results of the training process, we concluded that MLP is suitable for this kind of analysis, and therefore this network's training procedure is going to be described in this chapter.

The standard training procedure for MLP consists of two phases. In the first phase, a brief and fast spurt of BP takes place with a moderate training rate. This fulfills the "gross convergence" stage, and for some cases it is adequate, and it is not necessary to continue to the second phase, which is a lengthy run of conjugate gradient descent; this is a stronger algorithm that is more unlikely to encounter convergence issues.

The training data is by default 70% of the total data; therefore, the selection is made randomly by the machine. The training/selection and test data must be representative of the underlying mode. If the training data is not representative, then the model is compromised. The selection set is used to tune the parameters of the network and confirm that the network does not overfit. Using the test set, it is feasible to evaluate how the network performs in unseen data, in order to verify the predictive ability of the network (generalization). In order to perform a prediction, it is necessary to have a trained network; to know how good the network is, we need to check for the error function, which measures the proximity of the network predictions to the targets, and how much weight adjustment the training algorithm should apply in every iteration [63].

In the present study, the network topology was the combination of nine input nodes, one hidden layer with seven hidden nodes, and one output (9:1:7:1) (Figure 4). The output of the network training is the dependent variable, which is the erosion plots for year 1995 to 2008, and the inputs are the independent variables, which are the files with the number of residents in 2001 and 2011, the Corine land cover for the years 2000 and 2006, the number of households in 2001 and 2011, the vegetated and non-vegetated areas within the grey dune, and the files with the erosion plots in 1967–1980 and 1980–1995, as listed in Table 1.

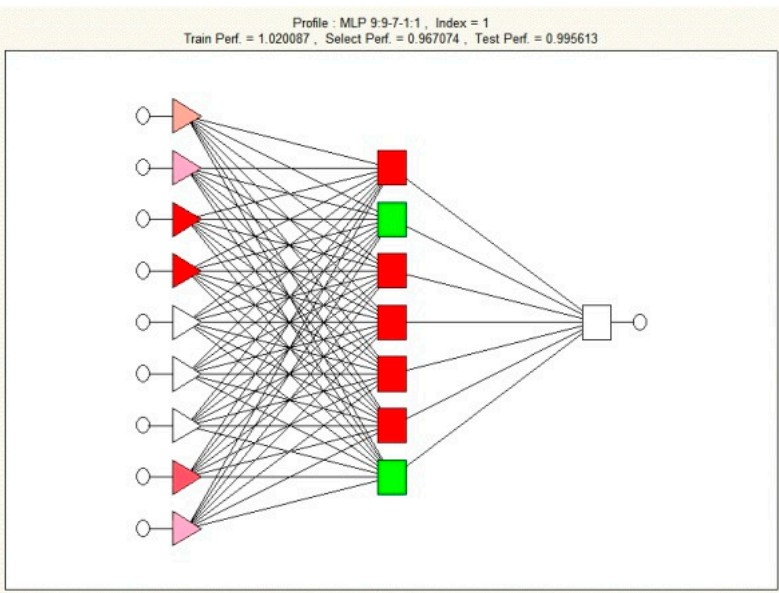

**Figure 4.** Network Topology 9:1:7:1. The train performance, selection performance, and test performance are similar, which means that the generalization ability of the network is satisfactory (authors).

ANNs always have one input and one output layer, and can have one or more hidden layers. ANNs that have one or more hidden layers are called multilayer networks, and therefore were given the name MLP [58]. The hidden layers have an important role in the machine learning process, as they are extracting the characteristics from the input layers and expressing them through the weights and the output layer. The number of hidden layers and hidden nodes in the hidden layers differ among

authors and study subjects [62]. For some, one hidden layer and two hidden nodes are enough for most of the problems [61]; for others, the numbers of nodes in the hidden layers should be arbitrary [64], and therefore a matter of empiricism and testing. Different network topologies and directions of signal propagation have been tried in order to conclude the final combination (9:2:6:3:1, 9:2:5:5:1, 9:2:4:3:1, 9:2:6:6:1). Following the network training, the sensitivity analysis of the inputs to the neural network was conducted.

The sensitivity analysis demonstrates the importance of the input variables by that specific neural network. It can give a better understanding of the usefulness of each variable. In other words, sensitivity analysis illustrates the variables that can be ignored in the following analyses, and the variables that must always be kept for prediction. To define the sensitivity analysis of a specific variable *v*, a set of test cases is used, and the network error is accumulated. Then, the network runs again, using the same cases and replacing the observed values of the *v* with the estimated value by the missing value procedure, and the network error is accumulated. If some information has been removed, some deterioration in error is logically expected. Sensitivity analysis is measured by comparing the ratio of the error with the missing value to the original error. The ratio is greater when the deterioration expected is greater, and therefore, the network is more sensitive to a specific input. When the ratio is one or less than one, the variable does not affect the performance of the network nor enhance it. When the ratio is calculated, then all of the variables are ranked in order, and the interpretation of the results is straightforward [62].

Once a network is fully trained, it is stabilized, which allows performing predictions on further samples that have been created under the same relations and processes as the original trained set [63,65], as cited in [66]. The input data values of one or more time series are used to predict the future values of the time series. During the training process, the network learns to recognize the patterns in the training signals. A training pattern has a fixed number of N lagged observations y_1,y_2, . . . ,y_N of the time series in the training set. For one step ahead, prediction of the ANN is through using n input nodes and N–n training patterns [67]. The training stops when the network finds the global minima, which is normally after iteration training tests with various initial weight values. After every training test, the results are evaluated and compared with the results of the previous training tests. Then, the best training test is selected, and the forecasting of future values follows. In this study, the network learns the patterns of the "area erosion plots" polygons from three different time series, and forecasts the erosion changes for the year 2021.

## 3. Results

After training the network several times, the results show that all of the variables are relevant for the study. Looking at the numbers of the variables' ratio on the different sensitivity analysis result tables (Table 2), there is no significant difference between the weights of the variables. This leads to no need for further network training, because it can cause noise in the model and during training. Finally, the network was trained with nine inputs, one hidden layer with seven hidden nodes, and one output (9:1:7:1) (Table 3). The variables of the number of residents and households seems to have a greater effect than the other variables in the areas that are vulnerable to erosion in the near future for the study area.

The results show that the anthropogenic variables, residents and households, are the main drive factors. In fact, the residents for 2001 and households for 2001 have greater effects than the residents and households for 2011. The reason for that is because there is a negative variation between 2001 and 2011 for both variables at Trafaria, and therefore, the variable that influences the output more is the oldest one.

The result of the predicted values for the network's output has been mapped (Figure 5) and classified in different levels of coastal erosion changes. In this study, "coastal erosion change" refers to the probability of a particular area within the study area to be eroded in the year of the prediction (2021) as a result of the anthropogenic pressure to the area. The values have been classified using the

quantile classification method into three classes ranging from a minimum value of −75 to a maximum value of 18. Thus, the break values for the three classes were three, four, and 18. Zero predicted values represent no changes. Different areas (polygons) with different colors, on a qualitative nominal scale, were created to show where low, medium, high, and no changes will take place in relation to the rest of the variables in the year 2021. The yellow color (predicted values −75 to three) indicates the areas that are affected less, the orange color shows the areas that are affected moderately (predicted value four), the red color indicates the areas that are highly affected (predicted values five to 18), and white shows the areas where there is no erosion change.

**Table 2.** Results of Sensitivity Analysis for Different Network Topologies (A, B, C, D, E). The rank shows which of the variables influence the output more, and has a range of one (high) to nine (low). The ratio shows how relevant the variables are for the study (authors).

| A | Sensitivity Analysis—network 9:2:6:3:1 | | | | | | | | |
|---|---|---|---|---|---|---|---|---|---|
| | ALOJ01_3 | ALOJ11_3 | CORNL_00 | CORNL06_ | ER6080_R | ER8090_R | VEG_RCID | RESID01_ | RESID11_ |
| Ratio | 1.000451 | 0.999655 | 0.999669 | 1.004549 | 1.017696 | 1.000097 | 1.007850 | 1.005299 | 1.000919 |
| Rank | 6 | 9 | 8 | 4 | 1 | 7 | 2 | 3 | 5 |
| **B** | Sensitivity Analysis—network 9:2:6:6:1 | | | | | | | | |
| | ALOJ01_3 | ALOJ11_3 | CORNL_00 | CORNL06_ | ER6080_R | ER8090_R | VEG_RCID | RESID01_ | RESID11_ |
| Ratio | 1.047523 | 0.999605 | 1.002351 | 1.000591 | 1.018752 | 1.000064 | 1.011311 | 1.004844 | 0.999059 |
| Rank | 1 | 8 | 5 | 6 | 2 | 7 | 3 | 4 | 9 |
| **C** | Sensitivity Analysis—network 9:2:5:5:1 | | | | | | | | |
| | ALOJ01_3 | ALOJ11_3 | CORNL_00 | CORNL06_ | ER6080_R | ER8090_R | VEG_RCID | RESID01_ | RESID11_ |
| Ratio | 1.012291 | 1.001607 | 0.998228 | 1.001061 | 1.013966 | 0.999958 | 1.001469 | 1.198692 | 1.003025 |
| Rank | 3 | 5 | 9 | 7 | 2 | 8 | 6 | 1 | 4 |
| **D** | Sensitivity Analysis—network 9:2:4:3:1 | | | | | | | | |
| | ALOJ01_3 | ALOJ11_3 | CORNL_00 | CORNL06_ | ER6080_R | ER8090_R | VEG_RCID | RESID01_ | RESID11_ |
| Ratio | 1.000000 | 1.000000 | 1.000000 | 1.000000 | 1.000000 | 1.000000 | 1.000000 | 1.000000 | 1.000000 |
| Rank | 4 | 7 | 2 | 5 | 3 | 9 | 8 | 1 | 6 |
| **E** | Sensitivity Analysis—network 9:1:7:1 | | | | | | | | |
| | ALOJ01_3 | ALOJ11_3 | CORNL_00 | CORNL06_ | ER6080_R | ER8090_R | VEG_RCID | RESID01_ | RESID11_ |
| Ratio | 1.033375 | 1.002661 | 0.998286 | 1.019494 | 1.019418 | 1.000044 | 1.004505 | 1.262340 | 1.007049 |
| Rank | 2 | 7 | 9 | 3 | 4 | 8 | 6 | 1 | 5 |

**Table 3.** Performance of the Network Training (9:1:7:1) STATISTICA (authors).

| Index | Profile | Train Perf. | Select Perf. | Test Perf. | Train Error | Select Error | Test Error | Training/ Members | Inputs | Hidden (1) | Hidden (2) |
|---|---|---|---|---|---|---|---|---|---|---|---|
| 1 | MLP 9:9-7-1:1 | 1.020087 | 0.967074 | 0.995613 | 0.012059 | 0.034072 | 0.023768 | BP1b | 9 | 7 | 0 |

The areas that are more vulnerable to erosion changes are located within the grey dune, in the borders of the dune, and the fossil cliff, which is close to the river Tagus inlet, in the low-density urban part of the coastal zone, where anthropogenic movement appears, and finally, in agricultural areas. The high density urban areas of Costa da Caparica are less prone to erosion changes.

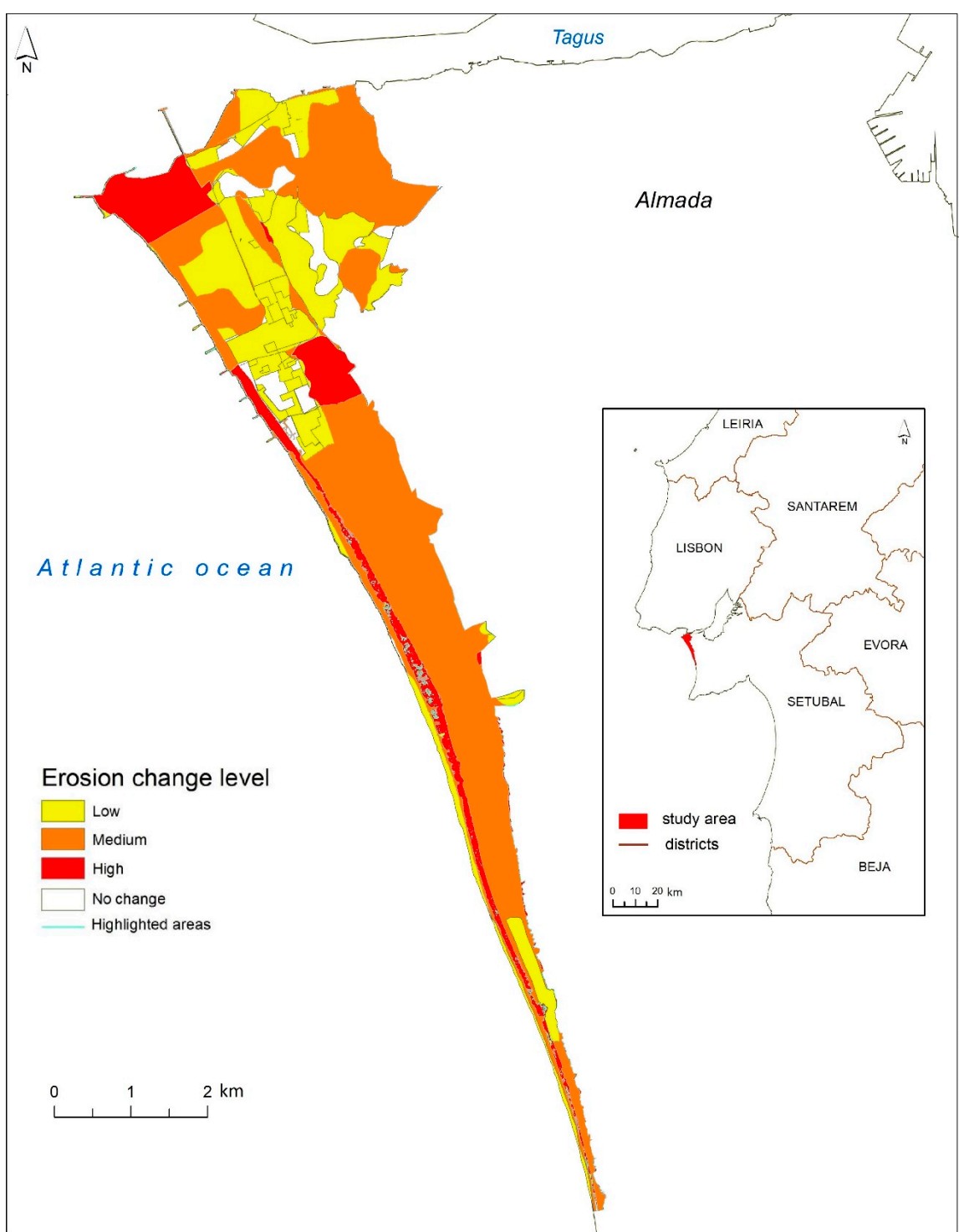

**Figure 5.** Map of the Areas Prone to Erosion Changes in 2021.

## 4. Conclusion and Discussion

The study area, as a sandy coastal zone system, is vulnerable to the dynamic balance of land and sea. Its morphology and species composition are the result of alterations of the shorelines (advance or retreat) over years caused by climatic changes, vegetation cover, sediment deposits, storm episodes, sea level variations, and anthropogenic activity [39]. In this study, the focus was on the prediction of areas that are prone to erosion affected by anthropogenic activity, testing two types of neural networks in combination with GIS.

The use of MLP technique for studying the coastal systems in Costa da Caparica was performed optimally. The model is fitted to reality, and suits the purpose of examining the impacts of anthropogenic activity in the coastal systems of the study area. The sensitivity analysis showed that the most important variables that affect the level of erosion change by 2021 are the number of residents (population) and the number of households (urbanization).

Here, it is important to clarify that population and urbanization are two different factors that, from one perspective, can combine to contribute to the cause of degradation of the natural environment, but conversely are not necessarily analogues. In other words, where there is an increase of urbanization, there is no implied growth of population, and vice versa. These are two different factors that affect the dynamics of the coastal system differently. Households/urbanization principally modify the hydrological and sedimentation regimes, as well as the nutrients and chemical pollutants dynamics. Urbanization creates impervious surfaces, resulting in the increase of surface runoff and the decrease of groundwater and waterway discharges. The quality of surface runoff is also changed, because it contains increased loads of sediment, nutrients, and pollutants. All of these affect the biota and physical environment. Population density affects the carrying capacity of the coastal zone. The population's change in size, composition, and distribution affect the coastal systems by altering the land uses and land cover of the area.

This methodology is different than other methods such as the Combined Coastal Vulnerability Index [68] because its offers a quantitative understanding of the coastal vulnerability; the areas are prone to erosion changes not only in the present, but also in the future (2021). Therefore, from the output map, as mentioned previously, we see that the areas with major erosion changes are situated within the grey dune, in the landward boundary between the beach–dune system and the fossil cliff, close to the river Tagus inlet, and in the part of the coastal zone that is more urbanized and has more human traffic. With an understanding of the coastal dynamics, the results seem accurate because there is a close relation between the erosion evidence and human activities such as excessive trampling, driving vehicles over the beach–dune system, and unregulated construction. Areas close to the river mouth are characterized by general instability, since they meander and change shape due to tidal inlets. Internal areas that appear to have major erosion changes are the areas that are within agriculture land use. Knowing that intensive agriculture causes soil degradation and therefore erosion makes the model fitted to reality.

At this point of the study, we are not able to measure the exact values of the predicted erosion. To be able to know that, it is necessary to use a classified erosion dependent variable instead of using the size of the eroded areas, as in the current study.

**Author Contributions:** Conceptualization, A.P. and P.M.; Methodology, A.P., P.M. and J.T.; Software, A.P. and P.M.; Formal Analysis, A.P., P.M. and J.T.; Data Curation; A.P. and P.M Writing—Original Draft Preparation, A.P. and P.M.; Writing—Review & Editing, A.P., P.M. and J.T.

**Acknowledgments:** We acknowledge the publication financing by Centre of Geographical Studies, Institute of Geography and Spatial Planning—Universidade de Lisboa and the anonymous reviewers that contributed to the improvement of this paper.

**Conflicts of Interest:** The authors declare no conflict of interest.

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
