# Peer review of "Combining Artificial Neural Networks and GIS Fundamentals for Coastal Erosion Prediction Modeling"

_sustainability, doi:10.3390/su11040975_

Round 1
Reviewer 1 Report
Overall comments:
Title: Combining Artificial Neural Networks and GIS fundamentals for Coastal Erosion Prediction Modelling
On overall it is a good paper with solid results. However, authors need revisions to improve the quality of the manuscript. The manuscript needs language consistency throughout the paper. It also needs language editing by native speaker.
Abstract
It’s good.
Introduction
The introduction is good and, however, several statements needs the references. While it can be expandable with more recent global examples. Please use the following examples.
1. Kantamaneni, K., Phillips, M., Thomas, T. and Jenkins, R., 2018. Assessing coastal vulnerability: Development of a combined physical and economic index. Ocean & Coastal Management, 158, pp.164-175.
2. Kantamaneni, K., Du, X., Aher, S. and Singh, R.M., 2017. Building blocks: A quantitative approach for evaluating coastal vulnerability. Water, 9(12), p.905.
Study Area
Very limited information has been provided by authors. This is not sufficient for full research papers. Authors should need to expand this section with study area physical and fiscal geography characteristics.
Methodology
The methodology is very innovative and unique. We need these type of methods. Combining Artificial Neural Networks and GIS fundamentals for Coastal Erosion Prediction is very interesting. However, the explanation of the methodology was so complicated to understand. I read several times to catch the methodological process. Authors need a simplified version for the methodology section. A lot of unnecessary information was provided such as an explanation about GIS.
Results and discussion
Without fully understanding of the methodology, It is not fair to give comments on the results.
For fig.5- please remove the title or heading from the top of the figure.
Conclusion
No conclusion was given by authors. Please provide this.
Author Response
Response to Reviewer 1 Comments
Introduction
The introduction is good and, however, several statements needs the references. While it can be expandable with more recent global examples. Please use the following examples.
1. Kantamaneni, K., Phillips, M., Thomas, T. and Jenkins, R., 2018. Assessing coastal vulnerability: Development of a combined physical and economic index. Ocean & Coastal Management, 158, pp.164-175.
2. Kantamaneni, K., Du, X., Aher, S. and Singh, R.M., 2017. Building blocks: A quantitative approach for evaluating coastal vulnerability. Water, 9(12), p.905.
Response: Introduction was re-written.
Study Area
Very limited information has been provided by authors. This is not sufficient for full research papers. Authors should need to expand this section with study area physical and fiscal geography characteristics
Response: Section was expanded.
Methodology
The methodology is very innovative and unique. We need these type of methods. Combining Artificial Neural Networks and GIS fundamentals for Coastal Erosion Prediction is very interesting. However, the explanation of the methodology was so complicated to understand. I read several times to catch the methodological process. Authors need a simplified version for the methodology section. A lot of unnecessary information was provided such as an explanation about GIS.
Response: Thank you for your kind and encouraging words. We have acted accordingly your suggestions. GIS subchapter has been removed and all chapter 2 (methods) has been re-written in order to make it more comprehensible and shorter as well.
Results and discussion
Without fully understanding of the methodology, It is not fair to give comments on the results. For fig.5- please remove the title or heading from the top of the figure.
Response: Methodology was re-written.
Conclusion
No conclusion was given by authors. Please provide this
Response: Concluding remarks were added in chapter Conclusion and Discussion.
Reviewer 2 Report
OUTLINE
The manuscript aims to propose a novel tool for the prediction of coastal erosion change “in the near future”. The joint using of GIS and Artificial Neural Network is proposed to predict the coastal erosion change by taking information from
I found that the manuscript is lacking for some crucial information and then I cannot recommend it for publication. However, I encourage the authors to address the following concerns and to submit a revised version of the manuscript that I will be glad to review again.
Main Concerns
#1
Through the manuscript, a “coastal erosion change” is often referred to. Nevertheless, I cannot find a clear definition. Indeed, the readers can clearly interpret the concept of "coastal erosion". Indeed, the manuscript is rather clear: “We can identify and measure erosion when the shorelines shift landwards and accretion when the shorelines shift seawards, comparing shorelines and coastlines in different time periods” (lines 37-38). On the other hand, what is considered as “coastal erosion change” is not declared. If a coastal area switches from erosion to accretion, can it be assessed as an “erosion change”? I ask the authors to define what they consider as “coastal erosion change”.
#2
The manuscript does not consider physical aspects of coastal erosion. The method relies on the use of anthropic pressure (i.e. resident population, number of households), land use and shoreline displacements. Sediment features (i.e. characteristics diameter), coastal system features (i.e. depth of closure), wave climate (i.e. the mean annual significant wave height nearshore), just to cite some of them, are not considered at all. From Fig. 5 I realize that a series of groins were deployed in the northern part of the considered coastal stretch. The presence of such defense structures are not considered in the analysis (and not described in the manuscript). The authors should clarify this aspect.
#3
The aim of the paper is not clear enough. The ANN output (see also my concern #1) is not clearly defined. Then, the readers cannot catch the details of the results. Just an example: how the “coastal erosion change” can be defined for internal areas (see Figure 5)?
#4
Sections 2.3.1, 2.3.2, 2.3.3 give information about ANNs that are almost known to the readers. I suggest shortening these sections by referring to the scientific literature. On the other hand, sections 2.4 and 3 are lacking details of the specific study. I suggest lengthening these sections. The last section (Discussion) does not seem to be strongly supported by the results. Concluding remarks are missing. I suggest the authors to strongly revise the structure of the manuscript.
#5 – Line 31-33
The manuscript states: “The land use and land cover (LULC) dynamics in CZs are changing due to the increase of inhabitation and urbanization largely due to the growth of the tourism industry and leisure activities as well as from sea level rise.” How “sea level rise” are changing the dynamics in CZs should be discussed more in details.
#6 – Lines 39-40
The manuscript states: “Identifying and monitoring the spatiotemporal shoreline position changes, it facilitates to recognize the way that the CZs react to human activities and natural processes.” I agree. However, natural processes are not considered in the study. Am I right?
#7 – Section “Introduction”
The manuscript proposed a novel tool to predict coastal erosion. The readers may be interested in how this prediction is performed currently. In other words, the introduction should describe also the physics-based method usually employed to manage coastal risk.
#8 – Line 108
The manuscript states: “The availability of sparse and partial data and the limitation of knowledge about the relationships between variables and factors is not an obstacle for ANNs’ application”. This principle is questionable. This is a question of philosophy. The concept could be rephrased as follows: “As we do not have enough data, and no knowledge of physics, we can use ANN”. I strongly disagree with this rationale. The ANNs application (basically a “black block” tool) can be employed only if the main physics is known, at least qualitatively. Then, the correct input to the ANN may be defined. The ANN role is to provide quantitative information. I ask the authors to clarify this aspect.
#9 – Line 311
The manuscript states “the results show that all the variables are relevant for the study.” I did not catch exactly the output of the ANN, however my question is “are the authors claiming that the considered variables suffice to predict the future coastal erosion?”. Or the results show that all the variable are (only) necessary. In my opinion (see also my concern #2), the considered variables may be necessary, but they do not suffice.
#10 – Lines 363-364
The manuscript states: “Dune erosion is caused naturally by the Aeolian activity (deflation) and the wind action.” Are “Aeloian activity” and “wind action” different? Moreover, it seems that the manuscript sees the “dune” as an autonomous component of the coastal (dynamic) system. I do not agree with this rationale. It is well known that the dune is a (vital) component of the coastal system and the sediments coming from or going to the dune is a measure of the resilience of coastal system against erosion. What do the authors think about that?
Author Response
Response to Reviewer 2 Comments
OUTLINE
Response: Thank you for all the comments and suggestions. We see them as very valid to improve the quality of our paper and increase our capability to communicate science. We also consider them as good tips for our further research articles.
MAIN CONCERNS
#1
Through the manuscript, a “coastal erosion change” is often referred to. Nevertheless, I cannot find a clear definition. Indeed, the readers can clearly interpret the concept of "coastal erosion". Indeed, the manuscript is rather clear: “We can identify and measure erosion when the shorelines shift landwards and accretion when the shorelines shift seawards, comparing shorelines and coastlines in different time periods” (lines 37-38). On the other hand, what is considered as “coastal erosion change” is not declared. If a coastal area switches from erosion to accretion, can it be assessed as an “erosion change”? I ask the authors to define what they consider as “coastal erosion change”.
Response #1: To create the erosion variable, we used accretion and erosion in different time periods based on different shorelines that have been created using DSAS (Digital Shoreline Analysis System) (Sousa, 2015) (see chapter 2.2. Data and chapter 2.4 Model implementation, 3. Results more information was added). In our paper “coastal erosion change” refers to the probability of a particular area within the study area to be eroded in the year of the prediction 2021 as a result of the anthropogenic pressure to the area. In this concept, the erosion change doesn’t include changes from erosion to accretion.
#2
The manuscript does not consider physical aspects of coastal erosion. The method relies on the use of anthropic pressure (i.e. resident population, number of households), land use and shoreline displacements. Sediment features (i.e. characteristics diameter), coastal system features (i.e. depth of closure), wave climate (i.e. the mean annual significant wave height nearshore), just to cite some of them, are not considered at all. From Fig. 5 I realize that a series of groins were deployed in the northern part of the considered coastal stretch. The presence of such defense structures are not considered in the analysis (and not described in the manuscript). The authors should clarify this aspect.
Response #2: The physical aspects of coastal erosion were considered in order to create the shorelines that were used from us to identify (and create) the erosion plots (depended variable). In this study, we don’t try to explain the erosion factors (the mechanical processes) but rather we try to identify the driving factors (which of the independent variables are triggering the futures changes) and the areas where it is more plausible erosion changes to occur. Regarding the comment about defense structure, information was added in chapter 2.1 Study Area.
#3
The aim of the paper is not clear enough. The ANN output (see also my concern #1) is not clearly defined. Then, the readers cannot catch the details of the results. Just an example: how the “coastal erosion change” can be defined for internal areas (see Figure 5)?
Response #3: The objectives of the paper were re-written in the introduction chapter in order to make it more explicit.
#4
Sections 2.3.1, 2.3.2, 2.3.3 give information about ANNs that are almost known to the readers. I suggest shortening these sections by referring to the scientific literature. On the other hand, sections 2.4 and 3 are lacking details of the specific study. I suggest lengthening these sections. The last section (Discussion) does not seem to be strongly supported by the results. Concluding remarks are missing. I suggest the authors to strongly revise the structure of the manuscript.
Response #4: We have re-written and pruned the text accordingly to the suggestions. Sections 2.3.1, 2.3.2, 2.3.3 have been re-written. Concluding remarks were added in chapter Conclusion and Discussion.
#5 – Line 31-33
The manuscript states: “The land use and land cover (LULC) dynamics in CZs are changing due to the increase of inhabitation and urbanization largely due to the growth of the tourism industry and leisure activities as well as from sea level rise.” How “sea level rise” are changing the dynamics in CZs should be discussed more in details.
Response #5: SLR-CZ-LULC-Population interactions were explained supported by extensive bibliographic references.
#6 – Lines 39-40
The manuscript states: “Identifying and monitoring the spatiotemporal shoreline position changes, it facilitates to recognize the way that the CZs react to human activities and natural processes.” I agree. However, natural processes are not considered in the study. Am I right?
Response #6: You are right in the sense that independent variables that could explain coastal erosion distribution over Costa da Caparica are not added to the model. But, as stated in the response to #2, our “area of erosion plots” results as a proof of evidence of the physical variables.
#7 – Section “Introduction”
The manuscript proposed a novel tool to predict coastal erosion. The readers may be interested in how this prediction is performed currently. In other words, the introduction should describe also the physics-based method usually employed to manage coastal risk.
Response #7: The other methods (physics-based) usually employ a linear analysis were variables are previous worked out and classified. Our model is non-linear and data-driven and therefore have the ability to incorporate uncertainty which is a fundamental property of complex systems such as coastal erosion. Another difference is that our model is more anthropogenic-based. Some explanations have been added to the Introduction, as requested in order to clarify the novelty of our model.
#8 – Line 108
The manuscript states: “The availability of sparse and partial data and the limitation of knowledge about the relationships between variables and factors is not an obstacle for ANNs’ application”. This principle is questionable. This is a question of philosophy. The concept could be rephrased as follows: “As we do not have enough data, and no knowledge of physics, we can use ANN”. I strongly disagree with this rationale. The ANNs application (basically a “black block” tool) can be employed only if the main physics is known, at least qualitatively. Then, the correct input to the ANN may be defined. The ANN role is to provide quantitative information. I ask the authors to clarify this aspect.
Response #8: We agree that the phrase needs further clarification. Thank you to bring it to our attention.
We agree that neural networks are a black box, although they’re not automatic or magic (Openshaw, S; Openshaw, C., 1997). In fact, they are designated as data-driven and empirical-based, which means that the data selection is crucial for the outcomes and that we need to test the variables and the neural parameters before decide for the best-fitted model. The neural network uses the observed data variables e.g. X1, X2, X3 and Y to “learn” how to relate the inputs to the output Y. It does it, without being restricted to linearity assumptions or hindered by knowledge of theory that may (or could) be wrong, biased or partial. It makes no assumptions about the statistical properties of the data nor about the units of measurement that are used since it can use numerical and categorical data.
That said, we agree that previous knowledge about the data, such as if there’s autocorrelation behavior or if there are enough data to test the hypothesis. Although, some authors have pointed out that for some of these applications, such as environmental studies, this may not be extremely critical, as it is for others such as seismic vigilance and monitoring (Balandrón, C., et al., 2012).
#9 – Line 311
The manuscript states “the results show that all the variables are relevant for the study.” I did not catch exactly the output of the ANN, however my question is “are the authors claiming that the considered variables suffice to predict the future coastal erosion?”. Or the results show that all the variables are (only) necessary. In my opinion (see also my concern #2), the considered variables may be necessary, but they do not suffice.
Response #9: From a scientific (empiricism-based) point of view, we can only be sure of which variables are or no sufficient if we perform several tests by adding more and different types of variables. Although regarding our results that we drawn by the data we use, these variables are relevant.
#10 – Lines 363-364
The manuscript states: “Dune erosion is caused naturally by the Aeolian activity (deflation) and the wind action.” Are “Aeloian activity” and “wind action” different? Moreover, it seems that the manuscript sees the “dune” as an autonomous component of the coastal (dynamic) system. I do not agree with this rationale. It is well known that the dune is a (vital) component of the coastal system and the sediments coming from or going to the dune is a measure of the resilience of coastal system against erosion. What do the authors think about that?
Response #10: The phrase was revised. We think that this misunderstanding is due to the lack of methodological explanation of how historical coastal erosion changes was determined. In fact, as we explain in the text now added to the methodology, this was achieved through the comparison of successive coastline delineation and this coastline proxy was the dune vegetation limit between the beach and the dune system.
We consider the coastal system, and particularly the beach-dune system of Costa da Caparica in a systemic and holistic perspective and long the text we give some examples of that:
Lines 19-23 - “In order to conduct sensitivity analysis on natural and social forces, as well as dynamic relations in the dune-beach system of the study area, two types of ANNs were tested on a GIS environment; Radial Basis Function (RBF) and Multilayer Perceptron (MLP). The GIS-ANN model helps to understand the factors which impact coastal erosion changes and the importance of having an Intelligent Environmental Decision Support System to address these risks.”
Lines 88-89 - “The coastal plain, due to its low topography and the fragility of the sand dune system, is vulnerable to flooding [11] and to erosion [14].”
Lines 168-171 - “…have been tested with different network topologies and training algorithms to predict the shoreline movements in the near future and to conduct sensitivity analyses on natural and social forces, and dynamic relations in the beach-dune-beach system of Costa da Caparica, Lisbon, Portugal.”
REFERENCES
Bayram, Savas; Ocal, M. Emin; Laptali Oral, Emel; Atis, C. Duran. Comparison of Multilayer perceptron (MLP) and Radial Basis Function (RBF) for Construction Cost Estimation: The Case of Turkey. J. Civ. Eng. Manag., 2016, 22(4), 480–490, doi:10.3846/13923730.2014.897988.
Baladrón, C.; Aguiar, J.M.; Calavia, L.; Carro, B.; Sánchez-Esguevillas, A.; Hernández, L. Performance Study of the Application of Artificial Neural Networks to the Completion and Prediction of Data Retrieved by Underwater Sensors. Sensors 2012, 12, 1468-1481, doi: 10.3390/s120201468
Yaïci, W.; Longo, M.; Entchev, E.; Foiadelli, F. Simulation Study on the Effect of Reduced Inputs of Artificial Neural Networks on the Predictive Performance of the Solar Energy System. Sustainability 2017, 9, 1382, doi: 10.3390/su9081382
Mezaal, M.R.; Pradhan, B.; Sameen, M.I.; Mohd Shafri, H.Z.; Yusoff, Z.M. Optimized Neural Architecture for Automatic Landslide Detection from High‐Resolution Airborne Laser Scanning Data. Appl. Sci. 2017, 7, 730, doi:10.3390/app7070730.
LeCunn, Y.; Bengio, Y.; Geoffrey, H. Deep Learning. Nature 2015, 521(7553), 436-444, doi:10.1038/nature14539.
Openshaw, S.; Openshaw, C. Artificial Intelligence in Geography, 1st ed.; John Wiley & Sons, Inc. New York, NY, USA, 1997, ISBN:0471969915.
Round 2
Reviewer 1 Report
Authors addressed all of my comments except adding following references.
Below mentioned references introduced new methods to evaluate coastal vulnerability and authors should refer in their introduction and discussion section.
1. Kantamaneni, K., Phillips, M., Thomas, T. and Jenkins, R., 2018. Assessing coastal vulnerability: Development of a combined physical and economic index. Ocean & Coastal Management, 158, pp.164-175.
2. Kantamaneni, K., Du, X., Aher, S. and Singh, R.M., 2017. Building blocks: A quantitative approach for evaluating coastal vulnerability. Water, 9(12), p.905.
Author Response
Response to Reviewer 1 Comments
Authors addressed all of my comments except adding following references.
Below mentioned references introduced new methods to evaluate coastal vulnerability and authors should refer in their introduction and discussion section.
1. Kantamaneni, K., Phillips, M., Thomas, T. and Jenkins, R., 2018. Assessing coastal vulnerability: Development of a combined physical and economic index. Ocean & Coastal Management, 158, pp.164-175.
2. Kantamaneni, K., Du, X., Aher, S. and Singh, R.M., 2017. Building blocks: A quantitative approach for evaluating coastal vulnerability. Water, 9(12), p.905.
Response: Thank you for reminding. The references have been added to Introduction and Discussion.
Reviewer 2 Report
OUTLINE
I thank the authors for having addressed my previous concerns. I found the manuscript improved.
However, in my opinion, some details deserve more attention. Indeed, the replies of the authors to my concerns were almost satisfactory, while not all the details were included in the revised version of the manuscript.
I will recommend the manuscript for publication once the authors address the following concerns.
CONCERNS
# 1 – Abstract – Line 15
The manuscript states: “The GIS-ANN model proves to be a powerful tool, as it analyses and provides the “where”, the “how” and “why” dynamics happen or will happen in the future.” However, the authors, in their reply #2 (I agree with) state “In this study, we don’t try to explain the erosion factors (the mechanical processes) but rather we try to identify the driving factors (which of the independent variables are triggering the futures changes) and the areas where it is more plausible erosion changes to occur.”
In my opinion, the proposed method aims to describe “where” and “why”, not “how” (from a physical point of view).
The authors should clarify this aspect.
#2 – Abstract – Line 25
There is a question mark (?). Perhaps a typo.
#3 – Introduction – Line 57
The manuscript states: “The aim of this paper is to provide a robust tool combining Geographic Information Systems (GIS) and Artificial Neural Networks (ANNs) in order to predict the areas, within the study area, more prone to erosion in the near future (year 2021).” I suggest, based on the authors’ replies, to add the following sentence “by taking into account the driving factors (i.e. driving independent variables) upon the erosion phenomena (i.e. the dependent variable).” Do the authors agree with me?
#4 – Introduction – Line 63
The manuscript states: “Understanding the shoreline changes, the relationships between the social and natural forces and forecasting future shoreline positions could be a vital tool for coastal management, decision-making, and land use planning and sustainable development [23] by collective behaviour of natural and human resources instead of individual behaviour of single categorical variables. Although, to really understand and explain the results highlighted by ANN, geography is the key, as local factors seem to be the root of such dynamics.”
Again, based on the authors’ replies to my first review I am not sure what the authors would express. Indeed, I understand that the manuscript does not “try to explain the erosion factors (the mechanical processes)” (authors’ reply #2). Then, the sentence does not seem to be appropriate in this context. I would suggest changing it to: “Understanding the relationships between the social and natural forces and forecasting future erosion trend could be a vital tool for coastal management, decision-making, and land use planning and sustainable development [23]. Although, to really understand and explain the results highlighted by ANN, the analysis of the mechanical processes is the key, as local factors seem to be the root of such dynamics.” Do the authors agree? If not, I suggest the authors to explicitly insert into the paper the main concept of their reply #2.
#4 – Study area – Line 108
It is rather strange to read two decimals when proposing an economic estimation. What the authors think about writing “between 50’000 and 1’000’000 euro”?
#5 – Table 1
Looking at the analyzed historical shoreline, I suggest changing the time references of “Area of erosion plots”. I mean, I suggest writing: “1967-1980” (instead of 60s-80s), “1980-1995” (instead of 80s-90s) and “1995-2008” (instead of “currently”).
#5 – Table 1
Table 1 should describe the units of each variable. Just for instance, when Land Use is considered, which is the way it is used as input of the proposed ANN?
#6 – Methods – Line 147
I suggest adding a section “2.3.1 GIS” (and then 2.3.2. Artificial Neural Network)
#7 – Methods – Figure 2
I suggest dropping Figure 2. Indeed, Figure 5 seems to be more appropriate and it suffices to make clear to the readers the concept of ANNs.
#8 – Methods – Line 208
Check the grammar (“output layers are further processing the data”).
#9 – Methods – Line 246
The manuscript states: “In the present study, the network topology was the combination of 9 input nodes, 1 hidden layer with 7 hidden nodes and 1 output (9:1:7:1) (Figure 5).”
I realize that the 9 input nodes are the variable shown in Table 1. However, Table 1 contains 10 variables. Then, I went to Table 2 and I realized that “Area of erosion plots – 1995-2008” is missing. Then, I argued that “Area of erosion plots – 1995-2008” is the output variable. Am I correct?
I ask the authors to clearly express which are the input nodes and which is the output node here (at least by clearly referring to Table 1).
#10 – Table 2
Ranks should be written as integer numbers.
#11 – Table 2 – Caption; Line 299
The caption of Table 2 reads “The rank shows which of the variables influence more the output and has a range of one (high) to nine (low).” In the text, the manuscript states: “The variables of the number of residents and households seems to have a greater effect than the other variables in the areas vulnerable to erosion in the near future for the study area.” Actually, the number of resident and households in 2001 (ranks equal to 1 and 2 respectively) have the greatest effect, while the number of resident and households in 2011 have the weakest effect (rank equal to 5 and 7 respectively).
The authors should comment on the ranks of each parameter by trying to explain why older data seem to be more important to future erosion trend with respect to new data.
#12 – Line 307
The manuscript states: “The result of the predicted values for the network’s output has been mapped (Figure 6) and classified in different levels of coastal erosion changes.” How the results (the output of the ANN) have been classified? The authors should declare the method used to classify the results.
#13 – Line 307
The manuscript states: “In this study “coastal erosion change” refers to the probability of a particular area within the study area to be eroded in the year of the prediction (2021) as a result of the anthropogenic pressure to the area.”.
I must be honest: this sentence confuses me.
Up to this point, I realize that the output of the ANN is the “Area of erosion plots – 1995-2008” (see concern #9), even if not explicitly declared. Thorough the manuscript, the output of the ANN is referred to as the “coastal erosion change”. Now, the manuscript refers to the probability of a given area to be eroded. How the probability was related to the output of the ANN? It is not clear enough.
Moreover, if the output of the ANN refers to the probability of a given area (within the study area) to be eroded in the year of the prediction, I cannot understand how internal areas could be eroded (see attached file).
The authors should clarify these aspects.

Author Response
Response to Reviewer 2 Comments
Point 1: Abstract – Line 15
The manuscript states: “The GIS-ANN model proves to be a powerful tool, as it analyses and provides the “where”, the “how” and “why” dynamics happen or will happen in the future.” However, the authors, in their reply #2 (I agree with) state “In this study, we don’t try to explain the erosion factors (the mechanical processes) but rather we try to identify the driving factors (which of the independent variables are triggering the futures changes) and the areas where it is more plausible erosion changes to occur.”
In my opinion, the proposed method aims to describe “where” and “why”, not “how” (from a physical point of view).
The authors should clarify this aspect.
Response 1: Thank you for the comment. We do agree with your view and we will change the text accordingly. In general terms, GIS can be seeing as a tool to address real world problems by providing the answers to quoted questions. Although, in our case study, the GIS-ANN model only reports on the “where” and “why”. So, thank you to call our attention.
Point 2: Abstract – Line 25
There is a question mark (?). Perhaps a typo.
Response 2: Thank you for bringing this to our attention. It has been removed.
Point 3:– Introduction – Line 57
“The aim of this paper is to provide a robust tool combining Geographic Information Systems (GIS) and Artificial Neural Networks (ANNs) in order to predict the areas, within the study area, more prone to erosion in the near future (year 2021).” I suggest, based on the authors’ replies, to add the following sentence “by taking into account the driving factors (i.e. driving independent variables) upon the erosion phenomena (i.e. the dependent variable).” Do the authors agree with me?
Response 3: We do understand the purpose of the suggestion and we do agree with you. The proposed sentence has been added to the text.
Point 4:– Introduction – Line 63
The manuscript states: “Understanding the shoreline changes, the relationships between the social and natural forces and forecasting future shoreline positions could be a vital tool for coastal management, decision-making, and land use planning and sustainable development [23] by collective behaviour of natural and human resources instead of individual behaviour of single categorical variables. Although, to really understand and explain the results highlighted by ANN, geography is the key, as local factors seem to be the root of such dynamics.”
Again, based on the authors’ replies to my first review I am not sure what the authors would express. Indeed, I understand that the manuscript does not “try to explain the erosion factors (the mechanical processes)” (authors’ reply #2). Then, the sentence does not seem to be appropriate in this context. I would suggest changing it to: “Understanding the relationships between the social and natural forces and forecasting future erosion trend could be a vital tool for coastal management, decision-making, and land use planning and sustainable development [23]. Although, to really understand and explain the results highlighted by ANN, the analysis of the mechanical processes is the key, as local factors seem to be the root of such dynamics.” Do the authors agree? If not, I suggest the authors to explicitly insert into the paper the main concept of their reply #2.
Response 4: We do thank you for the suggestion. We also agree with you, that the sentences are now more in line with the changes previous made.
Point 4: Study area – Line 108
It is rather strange to read two decimals when proposing an economic estimation. What the authors think about writing “between 50’000 and 1’000’000 euro”?
Response 4: Absolutely right. Thanks to bring it to our attention.
Point 5: Table 1
Looking at the analyzed historical shoreline, I suggest changing the time references of “Area of erosion plots”. I mean, I suggest writing: “1967-1980” (instead of 60s-80s), “1980-1995” (instead of 80s-90s) and “1995-2008” (instead of “currently”)
Response 5: We do agree. Changes have been made.
Point 5: Table 1
Table 1 should describe the units of each variable. Just for instance, when Land Use is considered, which is the way it is used as input of the proposed ANN?
Response 5: New column with the units of each variable added to the table. The Corine land Cover in level 1 has been used as one of the inputs for the study.
Point 6: Methods – Line 147
I suggest adding a section “2.3.1 GIS” (and then 2.3.2. Artificial Neural Network)
Response 6: We did have a GIS section on our pre-1st revision paper, but we have been advised to remove it, so we did.
Point 7:– Methods – Figure 2
I suggest dropping Figure 2. Indeed, Figure 5 seems to be more appropriate and it suffices to make clear to the readers the concept of ANNs.
Response 7: Figure 2 it’s a schematic illustration of Radial Basis Function (RBF) and figure 5, it’s a schematic representation of Multilayer Perceptron (MLP). They are topological similar, but they are two different neural networks. We consider keeping it in order to show the difference in topology between the two types of networks tested here.
Point 8: Methods – Line 208
Check the grammar (“output layers are further processing the data”).
Response 8: It has been revised.
Point 9: Methods – Line 246
The manuscript states: “In the present study, the network topology was the combination of 9 input nodes, 1 hidden layer with 7 hidden nodes and 1 output (9:1:7:1) (Figure 5).”
I realize that the 9 input nodes are the variable shown in Table 1. However, Table 1 contains 10 variables. Then, I went to Table 2 and I realized that “Area of erosion plots – 1995-2008” is missing. Then, I argued that “Area of erosion plots – 1995-2008” is the output variable. Am I correct?
I ask the authors to clearly express which are the input nodes and which is the output node here (at least by clearly referring to Table 1).
Response 9: The output mentioned in line 247 concerns the network training which is the dependent variable “Area of erosion plots – 1995-2008”. Information has been added to the text in section 2.4. Model Implementation
Point 10: – Table 2
Ranks should be written as integer numbers.
Response 10: They have been changed.
Point 11: Table 2 – Caption; Line 299
The caption of Table 2 reads “The rank shows which of the variables influence more the output and has a range of one (high) to nine (low).” In the text, the manuscript states: “The variables of the number of residents and households seems to have a greater effect than the other variables in the areas vulnerable to erosion in the near future for the study area.” Actually, the number of resident and households in 2001 (ranks equal to 1 and 2 respectively) have the greatest effect, while the number of resident and households in 2011 have the weakest effect (rank equal to 5 and 7 respectively).
The authors should comment on the ranks of each parameter by trying to explain why older data seem to be more important to future erosion trend with respect to new data.
Response 11: Indeed. Thank you to have noted. When there are no significant changes from time period t0 to time period t1 or there is a negative variation, which was the case, then t0 are the one the dominant. A brief explanation was added to the text.
Point 12: Line 307
The manuscript states: “The result of the predicted values for the network’s output has been mapped (Figure 6) and classified in different levels of coastal erosion changes.” How the results (the output of the ANN) have been classified? The authors should declare the method used to classify the results.
Response 12: Description about the classification has been added.
Point 13: Line 307
The manuscript states: “In this study “coastal erosion change” refers to the probability of a particular area within the study area to be eroded in the year of the prediction (2021) as a result of the anthropogenic pressure to the area.”.
I must be honest: this sentence confuses me.
Up to this point, I realize that the output of the ANN is the “Area of erosion plots – 1995-2008” (see concern #9), even if not explicitly declared. Thorough the manuscript, the output of the ANN is referred to as the “coastal erosion change”. Now, the manuscript refers to the probability of a given area to be eroded. How the probability was related to the output of the ANN? It is not clear enough.
Moreover, if the output of the ANN refers to the probability of a given area (within the study area) to be eroded in the year of the prediction, I cannot understand how internal areas could be eroded (see attached file).
The authors should clarify these aspects.
Response 13: The aim to this study is to provide a model which forecasts the erosion changes in Costa da Caparica, Lisbon, Portugal, for 2021. Therefore, the mapped output of the analysis is the prediction for 2021. The prediction is the probability of a given area to be eroded. Since the prediction is based on how the erosion plots have been changed between 1967-1980 and 1980-1995 the output is indeed erosion changes for 2021 (2008-1995=13 yrs, 2008+13= 2021).
Internal areas could be eroded since the variables that affect these changes for this study include also anthropogenic factors. Anthropogenic activity occurs not only in the dune but also in the internal areas which represent agricultural land (see line 369) and open fields. More explanation has been added.
Round 3
Reviewer 2 Report
I really thanks to the authors for having addressed all my concerns. Eventually, I found the paper improved and I can recommend it for publication.